# Cyanobacterial Peptides in Anticancer Therapy: A Comprehensive Review of Mechanisms, Clinical Advances, and Biotechnological Innovation

**DOI:** 10.3390/md23060233

**Published:** 2025-05-29

**Authors:** Heayyean Lee, Khuld Nihan, Yale Ryan Kwon

**Affiliations:** 1Plamica Labs, Batten Hall, 125 Western Ave, Allston, MA 02163, USA; yale.kwon@plamica.com; 2Department of Medical School, Jinnah Sindh Medical University, Rafiqui H.J. Shaheed Road, Karachi 75510, Pakistan; khuld.nihan1@gmail.com

**Keywords:** cyanobacterial peptides, anticancer, cyclic peptides, depsipeptides, antibody–drug conjugates, synthetic biology, heterologous expression

## Abstract

Cyanobacteria-derived peptides represent a promising class of anticancer agents due to their structural diversity and potent bioactivity. They exert cytotoxic effects through mechanisms including microtubule disruption, histone deacetylase inhibition, and apoptosis induction. Several peptides—most notably the dolastatin-derived auristatins—have achieved clinical success as cytotoxic payloads in antibody–drug conjugates (ADCs). However, challenges such as limited tumor selectivity, systemic toxicity, and production scalability remain barriers to broader application. Recent advances in targeted delivery technologies, combination therapy strategies, synthetic biology, and genome mining offer promising solutions. Emerging data from preclinical and clinical studies highlight their therapeutic potential, particularly in treatment-resistant cancers. In this review, we (i) summarize key cyanobacterial peptides and their molecular mechanisms of action, (ii) examine progress toward clinical translation, and (iii) explore biotechnological approaches enabling sustainable production and structural diversification. We also discuss future directions for enhancing specificity and the therapeutic index to fully exploit the potential of these marine-derived peptides in oncology.

## 1. Introduction

Cyanobacteria (blue-green algae) constitute an exceptional reservoir of bioactive secondary metabolites, producing a vast array of natural compounds including peptides, polyketides, and alkaloids. To date, over two thousand distinct cyanobacterial natural products have been identified, with many showing significant pharmacological activities such as antimicrobial, antiviral, and anticancer effects [1]. In their natural aquatic environments, these compounds usually serve as chemical defenses or toxins. However, when appropriately harnessed and chemically optimized, the same molecules can become valuable therapeutics. Cyanobacteria have thus emerged as prolific sources for novel anticancer drug discovery [2,3].

Several cyanobacterial peptides have already made a substantial impact on modern cancer therapy. A prominent example is dolastatin 10, a potent cytotoxic pentapeptide originally isolated from a sea hare that likely accumulates metabolites from *Symploca* cyanobacteria. Dolastatin 10 inspired the synthesis of auristatins, which are highly potent analogs now used as toxic payloads in multiple FDA-approved ADCs, including brentuximab vedotin, polatuzumab vedotin, enfortumab vedotin, and tisotumab vedotin [4]. In these ADCs, the antibody selectively delivers the auristatin to tumor cells, limiting systemic exposure. Similarly, curacin A, a cyclopropane-containing hybrid polyketide–peptide from marine *Lyngbya*, showed potent antimitotic activity by binding tubulin, but its development was hampered by limited supply and synthetic difficulty [5]. Likewise, cryptophycins—cyclic depsipeptides from *Nostoc*—exhibited extraordinary potency as microtubule depolymerizers, but clinical development was halted due to dose-limiting toxicities [6].

Inspired by these pioneering examples, researchers have intensified the exploration and development of cyanobacteria-derived peptides in oncology. These compounds often possess unique cyclic frameworks, unusual amino acids, or hybrid peptide–polyketide structures that confer high stability and strong affinity for cellular targets [7,8]. Furthermore, cyanobacterial peptides frequently engage novel molecular targets or pathways that differ from those of conventional chemotherapies, making them promising candidates for overcoming drug resistance or tackling traditionally “undruggable” targets [9,10].

Despite the growing interest and therapeutic promise of these natural peptides, substantial challenges impede their clinical translation. Key difficulties include obtaining adequate quantities from natural sources, the complexity of chemical synthesis, and inherent toxicities when used systemically. Recent advances in biotechnology—including metabolic engineering, heterologous biosynthesis, genome mining, and synthetic biology—offer powerful strategies to address these issues. These approaches enable sustainable production and structural optimization/diversification of cyanobacterial peptides [11,12]. This review comprehensively addresses the anticancer potential of cyanobacterial peptides, with a particular focus on cyclic peptides and peptide–polyketide hybrids. It summarizes their primary structural classes and mechanisms of action, outlines current clinical progress, and highlights biotechnological advances enabling sustainable production and structural diversification. Finally, current challenges and future perspectives are discussed, emphasizing strategies for maximizing the therapeutic impact of cyanobacteria-derived peptides in oncology.

## 2. Anticancer Peptides from Cyanobacteria: Classes and Examples

### 2.1. Cyclic Depsipeptides and Lipopeptides

Cyclic peptides (including depsipeptides, which contain ester linkages) are among the most prevalent cyanobacterial metabolites with anticancer activity. Their head-to-tail cyclized backbones and incorporation of exotic amino acids often confer resistance to proteolysis and high affinity for intracellular targets [13]. A well-known family in this class is microcystins—toxic cyclic heptapeptides produced by freshwater genera like *Microcystis*. Microcystin-LR and related congeners are notorious liver toxins that inhibit serine/threonine protein phosphatases PP1 and PP2A, causing potent cytotoxicity in hepatocytes. In cancer research, microcystins have gained interest for their ability to kill tumor cells that express organic-anion-transporting polypeptides (OATPs), which actively import the peptide. In vitro, sub-nanomolar concentrations of microcystin-LR induced rapid cell death in HeLa cervical cancer cells engineered to express human OATP1B1/1B3, via PP2A inhibition [14,15]. This finding suggests microcystins could be exploited as selective anticancer agents if delivered specifically to OATP-positive tumor cells. For example, researchers are developing tumor-targeted microcystin conjugates (linking the toxin to tumor-seeking carriers like antibodies or nanoparticles) to treat OATP-expressing malignancies such as liver and pancreatic cancers [14,16]. While native microcystins are far too toxic for systemic use, they exemplify how a cyanobacterial “toxin” can be repurposed as a targeted warhead against cancer when suitable delivery mechanisms are in place.

Marine cyanobacteria have yielded a plethora of cyclic depsipeptides and lipopeptides with antiproliferative effects. One such group is laxaphycins; originally isolated from *Anabaena*, laxaphycins A and B are cyclic lipopeptides that act synergistically to kill cells. Recent studies show that certain B-type laxaphycins have cytotoxic activity against human neuroblastoma cells, apparently by disrupting cellular membranes and possibly mitochondrial function, although further research is needed to fully elucidate their targets [17]. Another promising molecule is somocystinamide A (ScA), a dimeric cyclic lipopeptide from a *Lyngbya* assemblage [18]. ScA has demonstrated sub-nanomolar antiproliferative potency against endothelial and cancer cells by triggering extrinsic apoptosis. Specifically, ScA activates caspase-8, leading to apoptotic cell death even in tumors lacking caspase-8, in part through effects on the tumor vasculature [19]. In vivo, a single low dose of ScA in a mouse model inhibited the growth of both caspase-8-positive and -negative tumors by targeting the tumor blood vessels. This unique dual mode of action—inducing cancer cell death while simultaneously disrupting tumor angiogenesis—makes somocystinamide A an intriguing lead for further development. It could potentially treat caspase-8-deficient malignancies or be used in co-therapy to normalize tumor vasculature [19].

Cyanobacterial cyclic peptides often interfere with the cytoskeletal architecture of cancer cells. For example, grassypeptolides and lyngbyabellins are cytotoxic cyclic depsipeptides from marine *Lyngbya* species known to bind components of the cytoskeleton, causing filament collapse, cell cycle arrest, and apoptosis in cancer cells [20,21,22]. Curacin A, though technically a hybrid polyketide–peptide, is often discussed alongside these cyclic peptides due to a similar mechanism: it binds to the colchicine site on β-tubulin, preventing microtubule polymerization. Curacin A’s inhibition of mitotic spindle formation halts the division of rapidly proliferating cells [6,23]. Similarly, cryptophycin-52 binds tubulin but, in contrast to curacin, destabilizes microtubules, leading to their depolymerization [24]. Cryptophycin-52 demonstrated remarkable cytotoxicity toward tumor cells, with an IC_50_ in the low picomolar range. Although cryptophycin-52 advanced to Phase II trials, it was discontinued due to cumulative neurotoxicity [6]. Nonetheless, its potent mechanism inspired efforts to develop safer analogs and even ADCs using cryptophycin derivatives as payloads [25]. These examples underscore the potency of cyanobacterial cyclic peptides as mitotic inhibitors—comparable to plant-derived tubulin drugs like vincristine and paclitaxel—but with novel scaffolds that could be harnessed in new delivery formats.

Several other cyanobacterial cyclic peptides exert anticancer effects by inhibiting key enzymes in cancer cells. Largazole, a 16-membered cyclic depsipeptide from a marine *Symploca* strain, is a potent class I HDAC inhibitor [26,27]. Largazole acts as a prodrug; its thioester side chain easily enters cells and is then reduced to a free thiol, which chelates Zn^2+^ in the HDAC active site. This leads to the accumulation of acetylated histones and induces growth arrest and differentiation in cancer cells [28]. In colon cancer models, largazole showed strong antiproliferative effects and even in vivo tumor suppression, with selectivity for HDAC1/2/3 over HDAC6 [29]. Efforts are underway to optimize largazole analogs for improved drug-like properties [30]. Another noteworthy example is the carmaphycins, a pair of peptide epoxyketones from a marine *Symploca* that irreversibly inhibit the proteasome. Carmaphycin A and B carry an α,β-epoxyketone warhead identical to that of the proteasome inhibitor carfilzomib, an approved multiple myeloma drug [31]. While the carmaphycins were initially reported for antiparasitic activity [32], recent structure–activity studies have shown that suitably optimized carmaphycin analogs can potently block the human 20S proteasome [31,33,34]. Proteasome inhibition is a proven strategy in oncology, so these cyanobacterial peptides provide a template for new proteasome-targeting anticancer agents, potentially with different selectivity profiles or tissue distribution compared to existing drugs.

### 2.2. Peptide–Polyketide Hybrids

Cyanobacteria commonly produce “hybrid” metabolites via mixed polyketide synthase (PKS) and nonribosomal peptide synthetase (NRPS) pathways [8]. These hybrid peptide–polyketide natural products often have complex macrocyclic structures ideal for engaging challenging biological targets. We have already touched on some PKS–NRPS hybrids above (e.g., curacin A and cryptophycin). Another prominent example is the apratoxins, a family of cyclic depsipeptide hybrids from the marine cyanobacterium *Moorea bouillonii*. Apratoxin A and its analogs uniquely target the co-translational translocation of proteins into the endoplasmic reticulum. They bind to the Sec61 translocon channel, blocking nascent polypeptides from entering the ER lumen [35,36]. Cancer cells treated with apratoxins experience a broad halt in the production of secretory and membrane proteins, including many receptor tyrosine kinases and growth factors essential for tumor growth. This results in cell cycle arrest and apoptosis. Apratoxin S4, a modern analog, downregulates oncogenic receptors such as EGFR and MET through this mechanism, resulting in strong antiproliferative effects. Notably, apratoxin S4 demonstrated pancreas-targeted activity in a mouse model of pancreatic cancer, suggesting it concentrates its effect in pancreatic tissue [37,38]. Apratoxins exemplify how cyanobacterial hybrid compounds can engage non-classical targets, such as the Sec61 protein translocation machinery, which are often inaccessible to conventional drugs, thereby opening new therapeutic strategies. Efforts to refine apratoxin analogs for cancer therapy are ongoing, including semisynthetic modifications to enhance safety profiles.

Another compelling hybrid compound is coibamide A, a macrocyclic NRPS-PKS product from a *Panamanian Leptolyngbya* cyanobacterium. Coibamide A induces apoptosis in various cancer cell lines through a multi-faceted mechanism; it activates caspase-dependent pathways, downregulates certain nutrient receptor proteins on the cell surface, and can trigger autophagy. Recent transcriptomic and genetic studies revealed that coibamide A likely targets the Sec61 translocon as well, thereby preventing proper synthesis of certain membrane proteins [39]. Additionally, coibamide A disrupts mTOR signaling and induces ER stress in tumor cells, contributing to its cytotoxic effects [40]. In vivo studies have demonstrated strong suppression of angiogenesis and tumor growth in mouse models [41]. Despite remaining in preclinical development, its unique activity against treatment-resistant cancers such as glioblastoma and triple-negative breast cancer makes it a compelling candidate for further investigation. Owing to its distinct combination of mechanisms, coibamide A may also serve as a valuable molecular probe for identifying novel druggable pathways in cancer [42].

Lastly, symplostatin 1 is essentially a structural homolog of dolastatin 10 isolated from *Symploca*. Symplostatin 1, along with synthetic dolastatin analogs, targets tubulin polymerization in the same manner as dolastatin 10 [4]. Although symplostatin 1 itself did not advance as an independent drug lead, the entire dolastatin/symplostatin class has had an outsized impact via the auristatin derivatives used as ADC payloads. As mentioned above, multiple FDA-approved anticancer ADCs currently employ monomethyl auristatin E or F as the cytotoxic payload. These ADCs are used to treat a range of cancers—from lymphomas to solid tumors like urothelial carcinoma and cervical cancer [43,44]. The success of the auristatins validates the strategy of pairing a highly toxic cyanobacterial peptide with a targeted delivery system to achieve clinical efficacy. It also showcases how a single cyanobacterial scaffold can spawn numerous therapeutic agents when optimized and delivered appropriately. Table 1 provides an overview of selected examples categorized by structure and mechanism.

## 3. Molecular Mechanisms of Action

Cyanobacterial peptides exert anticancer effects through a diverse array of molecular mechanisms. These mechanisms range from physical disruption of cellular structures to biochemical modulation of signaling pathways and gene expression. This breadth of modes of action is an important advantage, as it means cyanobacterial compounds can be matched to specific tumor vulnerabilities or combined to achieve synergistic killing. Below, we outline key mechanistic categories with examples. An overview of these mechanistic categories and their effects on cancer cells is illustrated in Figure 1.

### 3.1. Disruption of Microtubules and Actin (Mitotic Arrest)

A significant number of cyanobacterial peptides function as antimitotic agents, interfering with the microtubule or actin cytoskeleton required for cell division [45]. Many such compounds like dolastatin 10, dolastatin 15, curacin A, and cryptophycins bind tubulin and interfere with microtubule assembly or stability, thereby preventing formation of a functional mitotic spindle [6,23,24]. This blockade of microtubule dynamics causes cells to arrest in mitosis (G_2_/M phase) and subsequently undergo apoptosis [46]. Notably, because these peptides often bind tubulin at uncommon sites (e.g., dolastatin 10/15 at the vinca domain or curacin A at the colchicine site), they can overcome certain resistance mechanisms to conventional tubulin drugs (such as β-tubulin mutations or drug efflux) [6,24]. Overall, tubulin-targeting cyanobacterial peptides efficiently halt the proliferation of rapidly dividing cancer cells.

In addition to microtubule-directed agents, a smaller number of cyanobacterial peptides disrupt the actin cytoskeleton. For example, lyngbyabellins and grassypeptolides bind to actin filaments and destabilize them, impairing processes like cell division and migration [20,21,22]. This actin-depolymerizing activity can inhibit cancer cell motility, potentially reducing metastatic spread. Because actin is abundant in normal cells, such agents are less common as systemic drugs; however, localized or targeted delivery might enable their use against tumors, particularly to prevent metastasis which relies on actin-driven cell movement.

### 3.2. Induction of Apoptosis (Intrinsic and Extrinsic Pathways)

Virtually all anticancer cyanobacterial peptides ultimately trigger apoptosis in tumor cells. Some compounds preferentially activate the extrinsic (death receptor-mediated) pathway of apoptosis, for instance by directly activating caspase-8 [19,39]. Somocystinamide A (ScA) illustrates this mechanism; it was shown to require caspase-8 for its full cytotoxic effect, underlining the importance of extrinsic apoptosis in its activity [19,47]. Such extrinsic pathway activators are especially valuable for killing cancer cells that have impaired intrinsic apoptotic machinery (for example, due to high Bcl-2 levels or loss of p53), since they bypass the mitochondrial route of cell death. Conversely, other peptides predominantly engage the intrinsic (mitochondrial-mediated) pathway. This often occurs indirectly through severe intracellular stress—such as extensive DNA damage, endoplasmic reticulum stress, or proteasome inhibition (see Section 3.4 and Section 3.5)—or via direct damage to mitochondria. In many cases, even if a peptide’s primary target is not a classical apoptosis regulator, the downstream consequence of the cellular damage is the activation of apoptosis. Thus, apoptosis is a common fate of cancer cells exposed to these compounds. Moreover, some cyanobacterial peptides can heighten a tumor cell’s susceptibility to apoptosis induced by other therapies. For example, an HDAC-inhibiting peptide may upregulate pro-apoptotic genes, thereby enhancing the effect of a second apoptotic stimulus [48]. In summary, whether by extrinsic or intrinsic routes, induction of programmed cell death is a central outcome of these peptides’ action against cancer cells.

### 3.3. Inhibition of Histone Deacetylases (Epigenetic Reprogramming)

Epigenetic modulation via histone deacetylase (HDAC) inhibition is a hallmark mechanism for certain cyanobacterial depsipeptides. Largazole, derived from a Symploca strain, is a potent class I HDAC inhibitor that functions as a molecular switch; once inside the cell it releases an active thiol, which binds the Zn^2+^ in the HDAC active site and inactivates the enzyme. This leads to the accumulation of acetylated histones and a more open chromatin structure, re-activating genes that were epigenetically silenced in cancer cells [27,49]. Notably, HDAC inhibition can restore the expression of tumor-suppressor genes and pro-apoptotic factors that had been turned off. Consistent with these effects, largazole induces cell cycle arrest and apoptosis in various cancer models [28,50]. It has also demonstrated in vivo efficacy, with treated tumors showing markedly increased histone acetylation levels [27]. Another cyanobacterial HDAC inhibitor, santacruzamate A, although less potent than largazole, underscores that marine cyanobacteria produce bioactive molecules capable of epigenetic regulation [51]. By reversing aberrant deacetylation, these compounds promote a more normal gene expression profile in cancer cells and effectively suppress tumor cell growth.

### 3.4. Inhibition of the Proteasome (Proteostasis Disruption)

Inhibition of the proteasome is another prominent anticancer mode of action employed by cyanobacterial metabolites. The proteasome is the cellular machinery responsible for degrading ubiquitinated proteins, and many cancers rely on heightened proteasome activity to cope with high protein turnover and to avoid proteotoxic stress. The carmaphycins, for example, are peptide epoxyketones from a Symploca cyanobacterium that irreversibly block the 20S proteasome’s proteolytic activity. These molecules share the same reactive pharmacophore as the FDA-approved drug carfilzomib, conferring them with proteasome-inhibitory potency on par with that clinical agent [34]. By inhibiting proteasomal degradation, such compounds cause an accumulation of polyubiquitinated and misfolded proteins inside cancer cells. The resulting protein overload triggers the unfolded protein response and other stress pathways; if the damage becomes overwhelming, the cell undergoes apoptosis due to proteotoxic stress and cell-cycle arrest. Mechanistically, proteasome inhibition stabilizes pro-apoptotic factors (which are normally broken down) and disrupts cell cycle regulators, tipping the balance towards cell death.

Recognizing the critical role of the proteasome in tumor cell survival, researchers have explored cyanobacterial proteasome inhibitors as targeted therapeutics. For instance, efforts are underway to conjugate carmaphycin analogs to tumor-seeking antibodies, selectively delivering these potent toxins to cancer cells. Overall, by impairing the cellular protein degradation machinery, cyanobacterial proteasome inhibitors efficiently induce apoptosis in cancer cells that are dependent on high proteasome activity. This underscores the proteasome as a valuable anticancer target exploited by marine-derived natural products [33,34].

### 3.5. Blockade of Protein Translocation (Sec61 Inhibition)

A particularly novel mechanism employed by certain marine cyanobacterial peptides is the blockade of Sec61-dependent protein translocation into the endoplasmic reticulum (ER). The Sec61 translocon is the channel through which nascent polypeptides destined for secretion or membrane insertion are fed into the ER lumen. Apratoxin A, isolated from marine *Lyngbya*/*Trichodesmium*, was the first natural product found to bind Sec61α and halt its translocation function [36]. By clogging the Sec61 channel, apratoxin A broadly prevents the co-translational import of secretory and membrane proteins. As a result, cancer cells exposed to apratoxin A cannot properly express many cell surface receptors and secreted growth factors, since these proteins fail to be translocated and processed. This leads to a rapid loss of pro-tumor signals and consequently produces antiproliferative and anti-angiogenic effects, effectively starving the tumor of external survival cues [37,38]. Additionally, the mislocalization of normally secreted proteins triggers ER stress and an unfolded protein response, which can culminate in apoptosis [31].

Another cyanobacterial compound, coibamide A, also targets the Sec61 translocon [52]. Coibamide A, a structurally distinct cyclic depsipeptide, similarly blocks the insertion of new proteins into the ER, causing a failure in membrane protein expression. In nutrient-deprived conditions, coibamide A has been shown to induce autophagy (an attempted survival response) followed by cell death, highlighting its potent activity against stressed tumor cells [39,53]. Photo-crosslinking studies confirmed Sec61α as the direct binding target of both apratoxin A and coibamide A, although interestingly these molecules appear to bind different pockets of the Sec61 channel complex [36,42]. In either case, the functional outcome is the same: a blockade of the protein translocation machinery. From a drug development perspective, this multi-target mechanism is compelling because it simultaneously disrupts numerous oncogenic pathways by preventing the proper folding and localization of a wide range of pro-tumor proteins. Indeed, optimized apratoxin analogs have shown potent anticancer activity in vivo while operating via this unique mode of action [38]. In summary, cyanobacterial Sec61 inhibitors effectively cut off the “supply lines” of cancer cells by blocking the secretion and surface display of many growth-promoting factors, leading to broad antitumor effects.

### 3.6. Protease Inhibition (Anti-Invasive and Cytotoxic Effects)

Cyanobacteria-derived peptides can also target proteolytic enzymes that are dysregulated in cancer. One notable example is gallinamide A, a modified tetrapeptide that acts as a potent and selective inhibitor of the cysteine protease cathepsin L. Cathepsin L is often overexpressed or secreted by cancer cells to degrade extracellular matrix components, facilitating tumor invasion and metastasis. Gallinamide A binds to cathepsin L with low-picomolar affinity, effectively blocking its activity [54]. By inhibiting this enzyme, gallinamide A can reduce the breakdown of tissue barriers, thereby hindering cancer cell invasion and spread. It may also interrupt pro-survival signaling pathways that depend on cathepsin L-mediated protein turnover. This mechanism highlights the therapeutic potential of targeting cysteine cathepsins (such as cathepsins L or B) to counter cancer progression [55,56]. While gallinamide A was initially explored for antiviral and antiparasitic effects [54,57], its anti-metastatic mechanism is highly relevant to oncology.

In addition to cysteine proteases, marine cyanobacteria produce inhibitors of various serine proteases implicated in cancer. Examples include varlaxins, kempopeptins, and micropeptins, which can target enzymes like elastase, trypsin, or urokinase plasminogen activator. By blocking these matrix-degrading proteases, such compounds may suppress tumor cell invasion and angiogenesis [58,59,60]. Although their direct anticancer efficacy is still under investigation, by limiting protease activity these peptides restrict the ability of cancer cells to remodel their surroundings. In some cases, interference with essential proteolytic processes can also induce tumor cell apoptosis. Overall, the inhibition of pathogenic proteases represents another way cyanobacterial peptides impede cancer progression, chiefly by reducing invasiveness and weakening the supportive tumor microenvironment.

### 3.7. Modulation of COX-2 and Inflammatory Pathways

Inflammation in the tumor microenvironment is a known driver of cancer progression, and cyanobacterial compounds that modulate pro-inflammatory enzymes can contribute to anticancer activity. The most notable example is C-phycocyanin, a blue phycobiliprotein from the edible cyanobacterium Spirulina, which has demonstrated selective inhibition of cyclooxygenase-2 (COX-2) [61,62]. COX-2 is an inducible enzyme that produces prostaglandins promoting inflammation, angiogenesis, and immune evasion in tumors [63]. Studies show that phycocyanin can downregulate COX-2 expression and activity in cancer cells. In triple-negative breast cancer cells, phycocyanin treatment led to a dose-dependent decrease in COX-2 at both the mRNA and protein levels, correlating with reduced PGE_2_ production [64]. COX-2 inhibition confers anticancer effects by reducing VEGF-driven angiogenesis, suppressing prostaglandin-mediated survival signaling to induce apoptosis, and limiting invasiveness. Phycocyanin-mediated COX-2 suppression has been linked to upregulation of E-cadherin, an epithelial marker inversely related to metastasis, and downregulation of pro-invasive markers [65]. In addition to phycocyanin, anti-inflammatory cyanobacterial compounds such as scytonemin—a pigment rather than a peptide—may exert anticancer or chemopreventive effects by targeting NF-κB or COX-2 pathways [66]. In the context of therapy, COX-2 inhibitors are known to synergize with radiotherapy and chemotherapy [67]. For example, phycocyanin has been reported as a radiosensitizer, enhancing radiation-induced cancer cell death by inhibiting COX-2 and thereby dampening the cell’s inflammatory stress response to radiation [68]. In summary, cyanobacterial products that attenuate tumor-promoting inflammation provide a complementary mechanism to direct cytotoxicity, effectively “softening” the tumor environment and making cancer cells more susceptible to apoptotic triggers.

## 4. Clinical Development Status of Cyanobacterial Peptides

Translating cyanobacterial peptides from discovery into clinically approved drugs involves overcoming hurdles of toxicity, selectivity, and production. Despite these challenges, several cyanobacteria-inspired compounds have entered clinical trials, and a few have achieved regulatory approval. Below, we review both the successes and setbacks in clinical development, as well as promising agents in trials.

### 4.1. Antibody–Drug Conjugates with Auristatin Payloads

The most successful clinical application of cyanobacterial compounds so far is the use of dolastatin 10 analogs as payloads in antibody–drug conjugates. ADCs enable targeted delivery of a cytotoxic peptide to cancer cells, thereby limiting systemic exposure [4].

Brentuximab vedotin (Adcetris) was the first such ADC, approved in 2011 for relapsed Hodgkin’s lymphoma and anaplastic large-cell lymphoma (ALCL). It consists of an anti-CD30 antibody linked to MMAE (monomethyl auristatin E). Upon binding to CD30 on lymphoma cells, the ADC is internalized and releases MMAE, which then induces mitotic arrest and apoptosis [69]. Brentuximab vedotin produced significantly improved outcomes in CD30^+^ lymphomas and is now a standard therapy.Polatuzumab vedotin (Polivy) targets CD79b on B-cell tumors and was approved in 2019 for relapsed or refractory diffuse large B-cell lymphoma (DLBCL) in combination with chemotherapy. In the pivotal phase II trial (GO29365), adding polatuzumab vedotin to bendamustine + rituximab significantly improved outcomes versus bendamustine + rituximab alone. Patients receiving the polatuzumab combination had a higher complete response rate (40% vs. 18%, *p* = 0.026) and superior median overall survival (12.4 vs. 4.7 months) [70].Enfortumab vedotin is an anti-Nectin-4 ADC carrying MMAE, granted accelerated approval in 2019 for metastatic urothelial carcinoma after platinum and PD-1/L1 therapy. In a single-arm phase II study (EV-201), enfortumab vedotin achieved a 44% overall response rate (12% complete responses) in heavily pretreated bladder cancer, with a median response duration of 7.6 months (FDA 2019). A phase III trial (EV-301) confirmed its benefit over chemotherapy; enfortumab vedotin monotherapy yielded an objective response rate of ~40% (vs. ~18% with chemo) and significantly prolonged median overall survival (12.9 vs. 9.0 months, HR 0.70, *p* ~0.001) in this setting [71]. This marked the first therapy to improve survival in post-immunotherapy bladder cancer. Enfortumab vedotin is also being explored in other Nectin-4-expressing tumors; for example, a cohort of patients with head and neck cancer showed a confirmed ~24% response rate on enfortumab vedotin [72], indicating activity beyond urothelial carcinoma.Tisotumab vedotin is an ADC against tissue factor, granted accelerated FDA approval in 2021 for recurrent or metastatic cervical cancer after chemotherapy. Approval was based on the phase II innovaTV 204 trial, which demonstrated a 24% objective response rate (7% complete responses), with a median response duration of 8.3 months in a refractory cervical cancer population [73]. A subsequent phase III trial (innovaTV 301) confirmed a clinical benefit over chemotherapy. In that randomized study, tisotumab vedotin improved median overall survival (11.5 vs. 9.5 months, HR 0.70, *p* = 0.004) and produced higher response rates (18% vs. 5%) than investigator’s choice chemo [74].Disitamab vedotin is a HER2-directed ADC (comprising an anti-HER2 antibody attached to MMAE) approved in China in 2021 for HER2-positive advanced gastric cancer, including tumors with low HER2 expression. In a pivotal single-arm phase II trial in patients with HER2-overexpressing gastric/gastroesophageal junction cancer who had failed ≥2 prior regimens, disitamab vedotin achieved a 24.8% objective response rate (95% CI 17.5–33.3%) [75]. Although the ORR was modest, some responses were durable, and median overall survival was ~7.9 months in this late-line setting. Beyond gastric cancer, disitamab vedotin has shown notable activity in HER2-positive urothelial carcinoma. A combined analysis of two phase II studies in advanced bladder cancer reported a confirmed ORR of ~50% with disitamab vedotin monotherapy [76] in patients refractory to standard chemotherapy. This high response rate, along with a manageable safety profile, highlights the promise of disitamab vedotin in HER2-expressing urothelial tumors and supports ongoing trials in these indications.Belantamab mafodotin is a B-cell maturation antigen (BCMA)-targeted ADC that received accelerated approval in 2020 for relapsed or refractory multiple myeloma after at least four prior therapies (including a proteasome inhibitor, an immunomodulatory drug, and an anti-CD38 antibody). The approval was driven by the phase II DREAMM-2 study, in which single-agent belantamab mafodotin showed an ~31% overall response rate in heavily pretreated myeloma patients [77]. Notably, a proportion of patients achieved responses lasting ≥6 months, introducing a novel mechanism (delivering MMAF to BCMA-expressing plasma cells). However, belantamab mafodotin was associated with corneal toxicity (keratopathy), necessitating careful eye monitoring and dose adjustments. Importantly, follow-up trials raised questions about its risk–benefit profile. The confirmatory phase III DREAMM-3 trial, comparing belantamab mafodotin to standard pomalidomide–dexamethasone, failed to show an improvement in progression-free survival. Despite a respectable response rate (~41% with belantamab in DREAMM-3), there was no significant PFS or overall survival advantage over standard therapy [78]. Consequently, in November 2022 the manufacturer voluntarily withdrew belantamab mafodotin from the US market due to insufficient efficacy in the confirmatory study. Ongoing trials are now exploring belantamab in combination regimens (e.g., with proteasome inhibitors or immunomodulators) to see if its benefit can be improved in earlier lines of myeloma treatment.

Collectively, these ADCs underscore the impact of cyanobacterial peptides in oncology; as of mid-2025, five FDA-approved cancer drugs on the market have been built around auristatin payloads. Each of these conjugates exploits the auristatin’s potent tubulin inhibition while the antibody confers tumor specificity. Ongoing trials are exploring these ADCs in combination with immune checkpoint inhibitors (for example, enfortumab vedotin + pembrolizumab in frontline bladder cancer has shown a very high response rate) and in earlier disease settings. The ADC modality continues to expand with new targets (e.g., an anti-CD25 MMAE-ADC for Hodgkin’s lymphoma is in trials) and new peptide payloads beyond auristatins [79,80]. Key regulatory approvals and clinical data for these ADCs are summarized in Table 2.

### 4.2. Clinical-Stage Investigational Agents (Phases I–III)

Beyond the approved drugs above, other cyanobacterial peptides or their analogs have entered clinical trials (2017–2025) for cancer therapy or supportive care. Here, we summarize notable examples:Glembatumumab vedotin is an ADC targeting glycoprotein NMB (gpNMB), a protein often overexpressed in triple-negative breast cancer (TNBC) and melanoma. In the phase 2b METRIC trial for metastatic TNBC, glembatumumab did not improve progression-free survival compared to chemotherapy (median 2.9 vs. 2.8 months, *p* = 0.97), failing to meet its primary endpoint [81]. While some tumor responses occurred, toxicity (notably rash and neutropenia) was significant, and the program was discontinued. Earlier phase II data in melanoma also showed only modest activity [82].Depatuxizumab mafodotin (ABT-414) is an EGFR-targeted ADC with an MMAF payload, developed for glioblastoma. Initial trials in EGFR-amplified gliomas showed some promise, but the phase III INTELLANCE-1 trial in newly diagnosed EGFR-amplified glioblastoma was negative. Depatux-M (with chemoradiation) did not improve survival versus standard chemoradiation plus placebo [83,84].Ladiratuzumab vedotin is an ADC targeting LIV-1 (a zinc transporter). It is being evaluated in TNBC and other solid tumors. An ongoing phase Ib/II trial in first-line TNBC combines ladiratuzumab vedotin with pembrolizumab. Preliminary results indicate manageable toxicity and evidence of activity in TNBC and other LIV-1–expressing tumors [85]. In a first-line TNBC cohort, the combination showed a preliminary ORR of about 33% in PD-L1^+ patients, supporting further development [85,86].Tasidotin (ILX-651) is a synthetic dolastatin-15 analog evaluated in advanced solid tumors. Phase I trials showed dose-limiting neutropenia at higher doses. Although no dramatic tumor regressions were seen, tasidotin did exhibit some anticancer activity. Notably, one melanoma patient achieved a complete response, and several patients had prolonged stable disease in early trials [87].Soblidotin (TZT-1027) is another dolastatin derivative (analog of dolastatin-10) that reached clinical trials with limited success. In a phase II study in refractory non-small cell lung cancer, soblidotin produced no objective tumor responses and a short median time to progression (~1.5 months). The trial concluded that soblidotin lacked meaningful anticancer activity in that setting, and further development for NSCLC was not pursued [88].Plitidepsin (Aplidin) is a cyclic depsipeptide originally isolated from a marine tunicate (sea squirt) but sometimes produced by a cyanobacterial symbiont. It has been tested in hematologic malignancies, particularly multiple myeloma. In the randomized phase III ADMYRE trial for relapsed/refractory myeloma, plitidepsin plus dexamethasone showed a significant improvement in progression-free survival compared to dexamethasone alone [89]. The combination also achieved a higher response rate (ORR ~13.8% vs. 1.7% with dex alone; *p* < 0.01) in this heavily pretreated population [89]. Interestingly, plitidepsin was repurposed during 2020–2021 as an antiviral agent against COVID-19 due to activity against SARS-CoV-2. In early 2024, a phase III trial in hospitalized COVID-19 patients showed faster viral clearance with plitidepsin [90]. While that is outside oncology, it highlights the broad potential of marine/cyanobacterial peptides in medicine. For cancer, plitidepsin is approved in Australia for myeloma and continues in trials elsewhere, indicating that such compounds can find niche clinical use [91].Spirulina (Arthrospira) and Phycocyanin are nutritional cyanobacteria that are also being evaluated as supportive care agents to mitigate side effects of cancer therapy. A 2019 randomized study (100 patients) found that dietary Spirulina during chemotherapy significantly reduced myelosuppressive toxicity. Patients who took Spirulina had higher post-chemo white blood cell and neutrophil counts and a lower rate of severe neutropenia compared to controls. They also experienced fewer dose delays and showed improved immune indicators (e.g., increased IgM and CD8^+^ T-cells) after therapy [61]. Meanwhile, phycocyanin (the antioxidant biliprotein from Spirulina) is being tested for preventing chemotherapy-induced peripheral neuropathy. The ongoing PHYCOCARE trial (phase II, NCT05025826) is evaluating oral phycocyanin vs. placebo in gastrointestinal cancer patients receiving oxaliplatin. The hypothesis is that phycocyanin’s ROS-scavenging properties will protect nerves from oxaliplatin neurotoxicity without compromising the anticancer efficacy of the chemotherapy [92]. Results are pending as of 2025.OKI-179 (bocodepsin) is an orally bioavailable prodrug analog of largazole (the marine cyanobacterial HDAC inhibitor). OKI-179 entered first-in-human trials in 2019 for advanced solid tumors. In a phase I dose-escalation study, OKI-179 was well tolerated, with manageable class-related toxicity (reversible thrombocytopenia as the dose-limiting toxicity) [93]. The drug showed dose-proportional exposure and robust HDAC target engagement at tolerated doses. Notably, OKI-179 induced histone acetylation in patient cells, confirming on-target activity. As of 2021, phase I results were encouraging, and an expansion phase Ib/II trial (“NAUTILUS”) launched to combine OKI-179 with a MEK inhibitor in RAS-mutant cancers. This represents one of the first HDAC inhibitor prodrugs derived from a marine cyanobacterial peptide to reach clinical testing.

To enhance the comprehensiveness of this section, Table 3 was added to provide an expanded overview of clinical-stage investigational agents inspired by cyanobacterial peptides. While several agents listed in Table 3 (e.g., glembatumumab vedotin, plitidepsin, OKI-179) are discussed in the text above, others—including lifastuzumab vedotin [94], PF-06263507 [95], telisotuzumab vedotin [96,97], mecbotamab vedotin [98,99], and ozuriftamab vedotin [100]—were included based on their use of auristatin payloads (MMAE or MMAF), which are synthetic analogs of dolastatin-10, a cyanobacterial peptide. These agents were not previously detailed in the main text but were incorporated to illustrate the broader clinical impact and structural versatility of cyanobacteria-derived cytotoxins in modern antibody–drug conjugate (ADC) development.

## 5. Biotechnological Strategies for Production and Optimization

A critical challenge in developing cyanobacterial peptides as drug candidates is achieving sufficient supply and consistency. Wild cyanobacteria often produce only trace amounts of a desired compound, and large-scale cultivation can be impractical [11,101]. Moreover, the complex structures of these peptides make total chemical synthesis difficult and costly. To address these limitations, researchers are employing various biotechnological approaches—including genomic mining, heterologous expression, metabolic engineering, synthetic biology, and scalable production systems—to enhance yield and enable structural modifications. Figure 2 illustrates an overview of these key approaches.

### 5.1. Genomic Mining and Activation of Silent Pathways

Genome sequencing has revealed that many cyanobacteria harbor numerous biosynthetic gene clusters (BGCs) that remain silent under standard laboratory conditions [101]. Strategies to unlock this hidden biosynthetic potential include the following:Heterologous Expression of BGCs: Large cyanobacterial BGCs can be cloned and expressed in more tractable hosts such as *E. coli*, *Anabaena* sp., or *Synechococcus elongatus*. For instance, the cryptomaldamide BGC from *Moorea* producens yielded high titers only when expressed in an *Anabaena* host, highlighting that expression can be host-dependent [102]. Similarly, the microginin BGC from *Microcystis* was expressed in *E. coli*, resulting in production of both expected and novel halogenated analogs (including variants not detected in the native strain) [52]. Greunke et al. used promoter refactoring in *E. coli* to enhance anabaenopeptin production from *Nostoc* by over 100-fold, demonstrating that heterologous expression coupled with synthetic regulatory elements can achieve scalable yields [103].Expression in Model Cyanobacteria: While *E. coli* and yeast are common heterologous hosts, model cyanobacteria like *S. elongatus* PCC 7942 offer a photosynthetic production platform that utilizes light and CO_2_. These hosts also naturally provide certain cofactors and chaperones that may be required for proper folding and activity of cyanobacterial enzymes. *S. elongatus* was able to support cryptomaldamide biosynthesis, providing an example of using a cyanobacterial chassis to express another cyanobacterium’s pathway (“self-compatible” expression system) [102].CRISPRa and Stress Induction: CRISPR-based activation (CRISPRa) involves using a catalytically inactive Cas9 (dCas9) fused to a transcriptional activator to upregulate silent gene clusters. Ke et al. applied this in Streptomyces, activating ten silent PKS/NRPS BGCs and uncovering 22 distinct metabolites [104]. While CRISPRa is still in early development for cyanobacteria, it holds promise for systematically accessing cryptic metabolomes [101]. Additionally, traditional elicitation approaches—such as subjecting cultures to UV light, nutrient limitation (e.g., iron starvation), or other stressors—have led to the activation of silent pathways, yielding compounds like cyanochelins and scytonemin. Overexpression of global regulatory genes has also proven effective in awakening latent biosynthetic activity in cyanobacteria [105,106,107].

### 5.2. Metabolic Engineering of Native Producers

Even when a BGC is naturally active, yields may remain too low for practical use. Metabolic engineering aims to reprogram the native cyanobacterial metabolism to channel more resources into target pathways and improve precursor availability for peptide biosynthesis.

Deletion of Competing Pathways: In *Synechococcus elongatus*, Choi et al. showed that CRISPR-Cas9 knockouts of competing central carbon metabolic genes (including those involved in glycolysis and phycobiliprotein biosynthesis) significantly increased production of isoprenoids. In particular, repression of the phycocyanin subunit gene (cpcB) diverted resources like ATP and amino acids toward heterologous product formation [108]. In another study, *Synechocystis* sp. PCC 6803 was engineered by deleting the shc gene, which encodes hopene cyclase—a key enzyme in hopanoid (triterpene) synthesis. This redirection of farnesyl diphosphate flux led to a marked increase in alternative triterpene accumulation [109]. These examples demonstrate that knocking out or downregulating competing pathways can free up cellular building blocks for the production of desired compounds.Amplifying Precursor Supply: For efficient peptide/polyketide biosynthesis, an ample supply of precursor metabolites (such as specific amino acids, malonyl-CoA, methylmalonyl-CoA, etc.) is essential. Roulet et al. boosted polyketide synthesis in *S. elongatus* by overexpressing enzymes that increase intracellular levels of malonyl-CoA and methylmalonyl-CoA [7]. Usai et al. combined genetic modifications with exogenous feeding of precursor molecules (e.g., 2-phenylethanol) in cyanobacteria, achieving a synergistic increase in final titers [110]. Such strategies can be adapted to NRPS pathways by supplying uncommon amino acid precursors or installing biosynthetic modules for unusual residues (ornithine, homoserine, etc.) to ensure the pathway is not limited by precursor availability.Promoter and Regulatory Engineering: Native BGCs are often under tight regulatory control and may only be weakly expressed. To overcome this, promoters within the pathway can be replaced with strong constitutive or inducible promoters. As mentioned, Greunke et al. did this in *E. coli* by refactoring the anabaenopeptin BGC, leading to >100-fold increase in production [103]. Choi et al. further used CRISPR interference (dCas12a-based repression) to simultaneously silence competing pathways while using synthetic promoters to boost the target pathway, amplifying terpenoid output [108]. Additionally, Leao et al. identified cryptic clusters in Moorea that were silent due to specific repressors [11]; by targeting those regulatory elements or co-expressing transcriptional activators, they were able to awaken those latent pathways.Optimizing Chassis Strains: The use of fast-growing and genetically tractable cyanobacterial strains such as *S. elongatus* UTEX 2973 has opened new possibilities for metabolic engineering. Knoot et al. showed that this strain (notable for its rapid growth) can serve as a high-biomass chassis for complex pathways, successfully producing hapalindole alkaloids after integration of the relevant BGC [111]. The ability of UTEX 2973 to quickly accumulate biomass and maintain high expression levels makes it ideal for large-scale biosynthetic applications where yield is critical.

### 5.3. Heterologous Expression in Non-Cyanobacterial Hosts

*Escherichia coli*: This bacterium is a popular host due to its fast growth, well-characterized genetics, and ease of manipulation. Several cyanobacterial peptides have been successfully produced in *E. coli*. For instance, the NRPS/PKS genes encoding hapalosin were cloned into an E. coli BAP1 strain, achieving approximately 45% of the native *Fischerella* yield [112]. Additionally, lyngbyatoxin A was produced in *E. coli* at the gram scale. *E. coli* has also been used to generate microcystin analogs by feeding alternative precursor amino acids to the NRPS machinery, enabling structure–activity relationship studies [113]. These successes illustrate *E. coli*’s versatility for heterologous expression, although it lacks some eukaryotic post-translational modification systems.Yeast (*Saccharomyces cerevisiae*): Yeast has emerged as a promising host for complex cyanobacterial pathways. In one study, a cyanobacterial NRPS-PKS pathway was reconstructed in *S. cerevisiae* to produce the sunscreen peptide shinorine. Deletion of a competing yeast pathway via CRISPR increased shinorine yield nearly 10-fold [114]. Yeast offers a eukaryotic expression environment that can support proper folding of large enzymatic complexes and provides subcellular compartmentalization, which can be advantageous for certain pathways. While yeast is not photosynthetic, its metabolic flexibility and its capacity to accommodate large DNA constructs, such as those introduced via yeast artificial chromosomes, make it a robust platform for the production of specialized metabolites [115].*Streptomyces*: These filamentous actinomycetes are renowned for producing antibiotics and are well suited for expressing large and complex cyanobacterial BGCs. Streptomyces species have high native expression of phosphopantetheinyl transferases (needed to activate NRPS/PKS enzymes) and abundant precursor pools. For example, *Streptomyces venezuelae* was used to express the 26 kb barbamide pathway from *Moorea* producens, resulting in production of the chlorinated metabolite [116]. More recent studies, using tools like bacterial artificial chromosomes and engineered Streptomyces strains, have achieved heterologous production of cyanobacterial compounds such as lyngbyatoxin A and teleocidins [12,117,118].

A particularly noteworthy achievement in heterologous expression was reported by Eusébio e al., who expressed a cryptic *Microcystis* BGC in *E. coli*. Not only did the host produce the expected microginin, but it also generated novel analogs not found in the native strain [52]. This metabolic interplay between the heterologous host and the introduced pathway highlights a key advantage of heterologous expression: the expansion of chemical diversity through host–pathway interactions. The physiology of the heterologous host can give rise to novel compound variants through differences in precursor availability or enzyme promiscuity, potentially leading to the discovery of bioactive molecules with enhanced properties.

### 5.4. Pathway Refactoring and Synthetic Biology

Synthetic biology enables modular redesign of biosynthetic pathways to improve productivity and diversify chemical structures. These strategies are especially important when natural pathways yield low titers or limited analog diversity.

Domain swapping and Module Editing: Rational swapping of domains in NRPS and PKS enzymes can change substrate specificity and produce new analogs. For example, Calcott et al. replaced adenylation domains in a Pseudomonas NRPS (for pyoverdine synthesis), which generated novel siderophores with altered amino acid composition [119]. In another example, Thong et al. used CRISPR-Cas9 genome editing to modify specificity-conferring domains in Streptomyces, reprogramming the enduracidin lipopeptide biosynthesis. The engineered strains produced new enduracidin analogs. The engineered strains produced new enduracidin analogs, confirming the success of module editing [120].Tailoring Enzyme Manipulation: Structural diversification can also be achieved by modifying tailoring enzymes in the pathway, such as methyltransferases or halogenases. For example, in the daptomycin lipopeptide pathway, knockout of the Glu12-specific methyltransferase yielded a desmethyl analog with altered pharmacological properties [121]. Similarly, Bradley et al. incorporated a tryptophan halogenase gene into a biosynthetic cluster, enabling the production of halogenated derivatives of a microbial alkaloid in vivo [122]. Such interventions expand the chemical space of known compounds by creating new analogs with potentially improved bioactivity or pharmacokinetics.Plug-and-Play Pathway Assembly: New cloning technologies now allow rapid refactoring and combinatorial assembly of entire biosynthetic pathways. Methods such as DiPaC (Direct Pathway Cloning) and TAR (transformation-associated recombination) facilitate one-step assembly of large biosynthetic gene clusters (BGCs) [123]. PCR-based refactoring of the complete erythromycin BGC in a single step. More recently, Ouyang et al. applied a similar strategy to clone and express a 16.8 kb radicicol biosynthetic pathway in E. coli, resulting in the production of novel analogs [123]. As further proof of concept, Basitta et al. reorganized the novobiocin antibiotic cluster into artificial operons using an assembly method (AGOS), leading to successful heterologous production [124].AI-Guided Design and Computational Modeling: Machine learning and computational tools trained on BGC databases can predict optimal engineering strategies. For instance, Kalkreuter et al. used molecular dynamics simulations to redesign acyltransferase domains in a modular PKS, expanding their substrate range [125]. Such in silico approaches increase the predictability and throughput of pathway engineering by highlighting beneficial mutations or domain swaps before laboratory implementation.

### 5.5. Scale-Up and Production Systems

Scaling up production is crucial for translating potent cyanobacterial metabolites into usable drugs. Two notable approaches are being pursued.

Photobioreactors: Engineered cyanobacteria can be grown in controlled photobioreactors at an industrial scale. For example, *Synechococcus* strains have been cultivated in 100–500 L photobioreactors using industrial flue gas as a CO_2_ source, successfully producing high-value compounds like squalene at scale [126]. Similar systems are being adapted for peptide production, offering a sustainable, sunlight-driven manufacturing process. Photobioreactors enable dense cultures under optimized light and nutrient conditions, potentially lowering costs for producing complex peptides compared to fermentation in the dark with expensive media [6,23,24].Cell-Free Biosynthesis: In vitro enzymatic systems can bypass living cells to produce natural products. By using purified NRPS/PKS enzymes or crude lysates, researchers have synthesized peptides in a cell-free manner. Notably, yields of approximately 30 mg/L have been achieved for certain model compounds, including cyclic dipeptides and the depsipeptide valinomycin, using *E. coli* lysate-based or two-stage cell-free systems [127,128]. Cell-free biosynthesis allows for precise control over reaction conditions and substrates, and it avoids issues of compound toxicity or metabolic regulation that occur in vivo. While currently used at small scales, advances in cell-free synthetic biology could enable on-demand production of complex cytotoxins by simply mixing the necessary enzymes and substrates in a reactor.

## 6. Current Limitations and Future Perspectives

### 6.1. Enhancing Specificity and Delivery

Many cyanobacterial peptides exhibit exceptional potency but suffer from insufficient tumor selectivity, leading to dose-limiting systemic toxicities. Innovations in targeted delivery—including improved ADC designs, tumor-specific nanoparticles, and cleavable small-molecule conjugates—offer potential solutions [24]. For instance, a highly cytotoxic cryptophycin analog showed picomolar anticancer efficacy but was abandoned in trials due to neurotoxic side effects, underscoring the need for targeted delivery. In a recent study, researchers conjugated a cryptophycin derivative to an anti-HER2 antibody, which yielded sub-nanomolar tumor cell killing and significant in vivo tumor suppression—outcomes unattainable with the free drug [129]. Future research should focus on optimizing linker stability and payload release kinetics in such conjugates, as well as identifying novel tumor-specific biomarkers to precisely target cyanobacterial warheads to cancer cells while sparing normal tissue.

### 6.2. Combination Therapies

Single-agent treatments often fail to cure advanced cancers due to tumor heterogeneity and adaptive resistance. Tumors can rapidly activate compensatory pathways under monotherapy, leading to outgrowth of resistant cell clones [130]. Combining cyanobacterial peptide toxins with other therapies offers considerable promise to overcome these escape mechanisms. In one study, adding a cyanobacterial peptide (microcystin-LR) to standard chemotherapy overcame an immunosuppressive tumor microenvironment and significantly enhanced anti-tumor efficacy in vivo compared to chemotherapy alone [99]. Similarly, co-inhibition of parallel signaling pathways has yielded synergistic effects; for example, dual therapy with a SHP2 inhibitor plus a KRAS-G12C inhibitor produced deeper and more durable tumor responses than either drug alone by abrogating feedback re-activation of wild-type RAS [131]. Moving forward, optimized clinical protocols are needed to validate such combinations, determine optimal dosing schedules, and minimize cumulative toxicity. The versatility of cyanobacterial agents (spanning various targets) makes them attractive partners in rational combination regimens.

### 6.3. Expanding Chemical Diversity

Despite extensive study, numerous cyanobacterial peptides remain undiscovered or underexplored. Genome mining surveys reveal that 185 sequenced cyanobacterial genomes collectively encode ~1817 biosynthetic gene clusters, of which only a fraction have known products [132]. Recent work underscores this untapped diversity. For example, a genome-informed discovery effort uncovered monchicamides A–K, a family of 11 new cyclic peptides from a marine Microcoleaceae cyanobacterium [133]. Advanced genomic mining, heterologous pathway expression, and synthetic biology techniques (Section 5) provide opportunities to unlock novel anticancer scaffolds that nature has “hidden”. A 2022 study cloned and expressed a cryptic microginin peptide pathway from *Microcystis*, which produced 12 new microginin variants featuring unusual amino acids [52]. Similarly, chemo-enzymatic synthesis has expanded known structures; one group generated and biologically evaluated a suite of cryptophycin analogs, identifying a styryl-cryptophycin derivative with sub-nanomolar cytotoxicity against cancer cells [134]. Continued bioprospecting in diverse aquatic environments, combined with innovative synthetic and computational methodologies, will be crucial to expanding the anticancer peptide arsenal. Each new structure provides not only a potential drug lead but also insights into structure–activity relationships that can guide further optimization.

### 6.4. Safety Profiling and Toxicology

Given their inherent toxicity, rigorous preclinical evaluation of cyanobacterial peptides is critical. Effective cancer treatments demand selective targeting of tumor cells with minimal harm to healthy tissues [135]. Early toxicological profiling in relevant models helps establish a therapeutic window and identify any organ-specific risks. Understanding detailed toxicokinetics, off-target effects, and dose-limiting toxicities (e.g., peripheral neuropathy from auristatins or hepatotoxicity from microcystins) is essential for safe clinical translation. For instance, prolonged low-dose exposure to monomethyl auristatin E in mice causes a sensory neuropathy with nerve fiber degeneration, mirroring the dose-limiting neurotoxicity seen in patients [136]. Likewise, oral dosing studies confirm that microcystin-LR is primarily a liver toxin that concentrates in hepatocytes via specific uptake transporters, leading to characteristic liver injury [137]. To mitigate such risks, advanced delivery technologies (e.g., prodrug formulations or nanoparticle encapsulation) are being explored to restrict the toxin to the tumor site and reduce systemic exposure. The utilization of cutting-edge preclinical models, including human organoid cultures and humanized mouse models, should become standard practice to better predict clinical safety and satisfy regulatory requirements. These models can more accurately recapitulate human drug responses than traditional rodent studies, helping to identify potential issues before entering the clinic.

### 6.5. Clinical Trials and Accessibility

Transitioning from bench to bedside also necessitates demonstrating clear clinical benefits beyond preclinical efficacy alone—including meaningful improvements in patient survival and quality of life. Notably, many oncology drugs that extend progression-free survival do not improve patient-reported quality of life, highlighting the need for new therapies that positively impact both longevity and well-being [138]. In designing clinical trials for cyanobacterial peptide-based agents, biomarker-guided patient selection will be key to identify those patients most likely to respond, thereby maximizing observed benefit and avoiding unnecessary toxicity in others. This approach has been exemplified by the approved ADCs that target defined tumor biomarkers (e.g., CD30 in lymphomas, Nectin-4 in urothelial carcinoma). Enfortumab vedotin, for example, was trialed specifically in patients with Nectin-4-positive tumors and achieved durable responses in metastatic urothelial cancer, leading to its approval in that biomarker-defined group [139]. Likewise, anti-CD30 ADC therapy has transformed outcomes in CD30-expressing Hodgkin’s lymphoma, significantly improving long-term survival in advanced disease [140]. These cases illustrate how matching a targeted cyanobacterial toxin conjugate with the right patient population can maximize clinical impact—ultimately improving not only survival but also quality of life in practice by avoiding ineffective treatment in non-responders.

## 7. Conclusions

Cyanobacterial peptides offer unique opportunities for anticancer therapy due to their distinctive mechanisms of action, including cytoskeletal disruption, epigenetic modulation, interference with proteostasis, and induction of apoptosis through targeted pathways. Clinically validated examples, such as auristatin-based antibody–drug conjugates (ADCs), illustrate the translational potential of these natural products when integrated with precise drug delivery systems. However, fully harnessing the therapeutic value of peptides derived from cyanobacteria requires overcoming several critical challenges, particularly enhancing tumor selectivity to reduce systemic toxicity and establishing reliable methods for the production of complex peptide structures. Recent biotechnological advances, including innovations in synthetic biology, heterologous biosynthesis, and genome mining, have significantly improved the availability of these compounds and enabled the development of analogs with enhanced pharmacological properties. Looking forward, interdisciplinary efforts should prioritize the refinement of targeted delivery strategies, the identification of synergistic combination therapies capable of preventing resistance and attacking tumors through multiple mechanisms, and the implementation of rigorous safety testing using advanced preclinical models that ensure an acceptable therapeutic index. Additionally, biomarker-guided clinical trials will be essential to match specific peptide-based agents with the patients most likely to benefit from them. Through continued innovation at the intersection of natural products chemistry, bioengineering, and clinical oncology, we can maximize the therapeutic impact of cyanobacterial peptides. These molecules, which originate from marine and freshwater microorganisms, have already shown the capacity to overcome certain forms of cancer resistance. With further development, they hold strong promise as novel and effective treatment options—particularly for cancers that are unresponsive to conventional therapies—thereby contributing meaningfully to the future landscape of cancer care.

## Figures and Tables

**Figure 1 marinedrugs-23-00233-f001:**
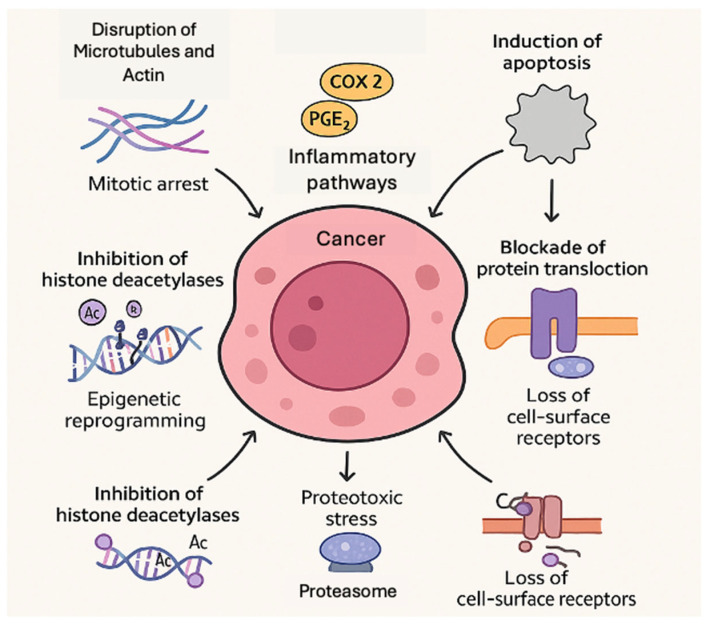
Overview of key anticancer mechanisms of cyanobacterial peptides acting on cancer cells.

**Figure 2 marinedrugs-23-00233-f002:**
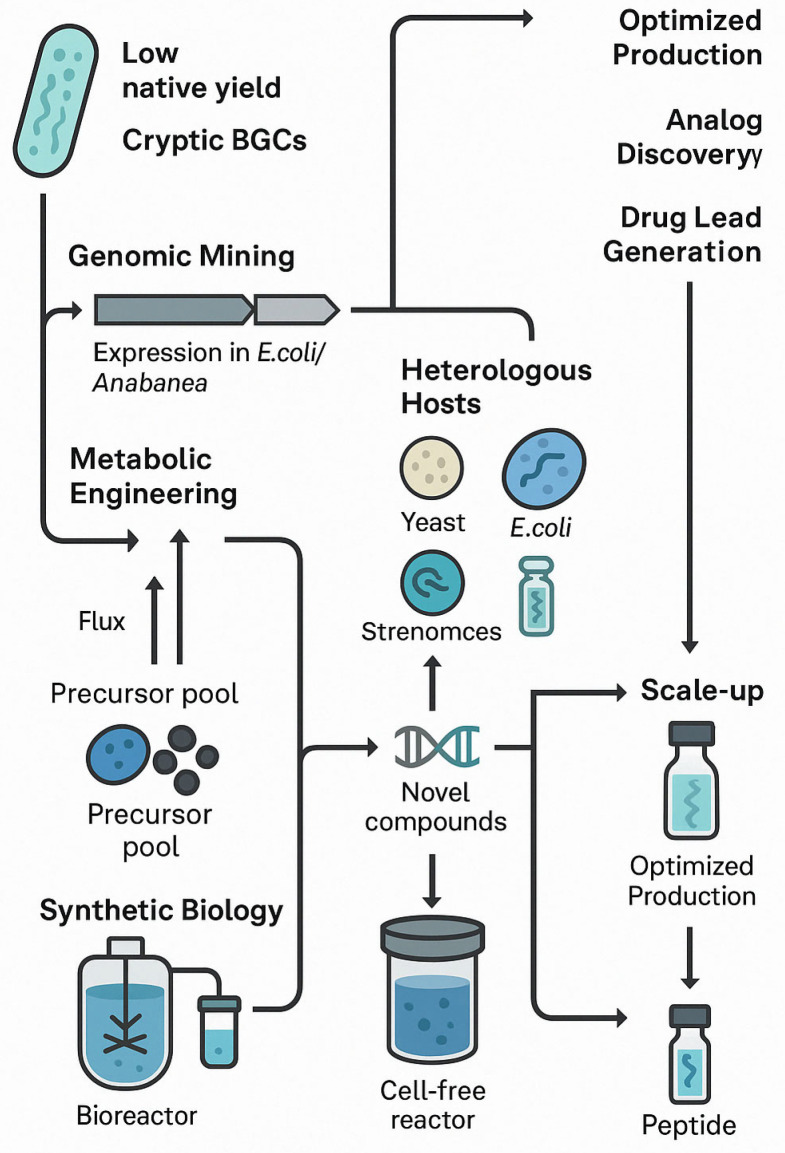
Schematic overview of biotechnological strategies for the optimized production of cyanobacterial peptides. The diagram illustrates key approaches, including genomic mining of cryptic biosynthetic gene clusters (BGCs); metabolic engineering to redirect precursor flux; heterologous expression in microbial hosts (e.g., E. coli, yeast, Streptomyces); synthetic biology-based pathway refactoring; and scale-up using photobioreactors or cell-free systems. These integrated strategies enable enhanced peptide yield, discovery of new analogs, and facilitate development of marine-derived drug leads.

**Table 1 marinedrugs-23-00233-t001:** Representative cyanobacterial peptides, their structural class, and primary anticancer mechanisms.

**Peptide (Source)**	**Structural Class**	**Primary Mechanisms**
Microcystin-LR (*Microcystis*)	Cyclic heptapeptide (depsipeptide)	Inhibits serine/threonine phosphatases PP1/PP2A; triggers rapid cytotoxicity
Laxaphycin A/B (*Anabaena*)	Cyclic lipopeptides	Disrupt cellular membranes (synergistic cytotoxic action)
Somocystinamide A (*Lyngbya*)	Dimeric cyclic lipopeptide	Induces extrinsic apoptosis (caspase-8 activation); anti-angiogenic (targets tumor vasculature)
Grassypeptolide A (*Lyngbya*)	Cyclic depsipeptide	Binds to actin filaments; destabilizes cytoskeleton (mitotic arrest and apoptosis)
Lyngbyabellin A (*Lyngbya*)	Cyclic depsipeptide	Binds to actin (some analogs bind tubulin); disrupts cytoskeletal dynamics
Dolastatin 10 (*Symploca*)	Linear pentapeptide (NRPS-derived)	Binds β-tubulin (vinca domain); prevents microtubule assembly (mitotic arrest)
Symplostatin 1 (*Symploca*)	Linear peptide (dolastatin analog)	Binds β-tubulin similar to dolastatin 10; precursor to auristatin analogs used in ADCs
Cryptophycin-52 (*Nostoc*)	Cyclic depsipeptide	Binds tubulin (distinct site); causes microtubule depolymerization (mitotic collapse)
Curacin A (*Lyngbya*)	Hybrid polyketide–peptide	Binds β-tubulin (colchicine site); inhibits microtubule polymerization (mitotic arrest)
Largazole (*Symploca/Caldora*)	Cyclic depsipeptide	Inhibits class I histone deacetylases (HDAC1/2/3); causes hyperacetylation of histones (epigenetic reprogramming)
Carmaphycin A/B ( *Symploca*)	Linear peptide (epoxyketone)	Irreversibly inhibits the proteasome (via epoxyketone warhead); blocks protein degradation, inducing cell death
Apratoxin A (*Moorea*/*Lyngbya*)	Cyclic depsipeptide (PKS-NRPS hybrid)	Blocks Sec61 translocon (inhibits co-translational protein translocation into ER); downregulates growth factor receptors
Coibamide A (*Leptolyngbya*)	Cyclic depsipeptide (PKS-NRPS hybrid)	Blocks Sec61 translocon; induces caspase-dependent apoptosis; disrupts mTOR signaling and angiogenesis

**Table 2 marinedrugs-23-00233-t002:** Cyanobacterial peptide-derived agents with regulatory approvals (2017–2025).

Agent (Type)	Indication	Regulatory Status	Key Efficacy/Safety Findings
Brentuximab vedotin (ADC, MMAE from dolastatin)	Hodgkin lymphoma, ALCL (CD30⁺ lymphomas)	Approved (US/EU 2011–12; expanded 2018+)	Improves PFS/OS in CD30⁺ lymphomas; first auristatin ADC to reach market/Common tox: peripheral neuropathy.
Polatuzumab vedotin (ADC, MMAE)	R/R DLBCL (with BR chemo)	Approved (FDA 2019; EU 2020)	+BR improved CR 40% vs. 18% and median OS 12.4 vs. 4.7 mo over BR alone. Priority review granted due to 58% lower risk of death/Notable tox: cytopenias, neuropathy.
Enfortumab vedotin (ADC, MMAE)	Metastatic urothelial carcinoma	Approved (FDA 2019; EU 2022)	ORR 44% (12% CR) in post-platinum/IO bladder cancer; confirmed OS benefit vs. chemo in Phase III/Tox: neuropathy, rash; rare serious hyperglycemia.
Tisotumab vedotin (ADC, MMAE)	Recurrent/metastatic cervical CA	Approved (FDA 2021)	ORR 24% (7% CR); median DOR 8.3 mo in refractory cervical cancer (single-arm Phase II); OS benefit vs. chemo (Phase III) /Tox: ocular (boxed warning for conjunctival/corneal injury).
Belantamab mafodotin (ADC, MMAF)	Relapsed multiple myeloma (BCMA-targeted)	Approved (FDA/EMA 2020); Withdrawn (US 2022)	~31% ORR in heavily pretreated myeloma (monotherapy) with some durable responses/Notable toxicity: corneal damage (keratopathy). Approval withdrawn after Phase III trial showed no PFS benefit over standard therapy.
Disitamab vedotin (ADC, MMAE)	HER2+ gastric/GEJ adenocarcinoma	Approved (China 2021)	~30% ORR in HER2 IHC 2+ or 3+ gastric cancer after 2+ lines (conditional approval). Showing activity in HER2-low tumors as well; Phase II completed (ORR ~25% gastric; ~50% bladder)/Key tox: nausea, marrow suppression, liver enzyme elevations.

Abbreviations: ALCL = anaplastic large-cell lymphoma; DLBCL = diffuse large B-cell lymphoma; BR = bendamustine + rituximab; ORR = objective response rate; CR = complete response; DOR = duration of response; PFS/OS = progression-free/overall survival.

**Table 3 marinedrugs-23-00233-t003:** Clinical-stage investigational agents derived from cyanobacterial peptides (Phases I–III, 2017–2025). This table expands the survey of candidates to include additional peptide-based agents inspired by cyanobacteria, with their targets, trial status, and outcomes.

Agent (Type)	Target	Indications	Phase	Status/Key Outcomes
Glembatumumab vedotin (ADC, MMAE payload)	gpNMB	Metastatic triple-negative breast cancer (TNBC); also tested in melanoma	Phase 2b (METRIC, NCT01997333)	No benefit over chemo in TNBC (median PFS 2.9 vs. 2.8 mo); significant toxicity (rash, neutropenia); discontinued after failing primary endpoint. Minimal activity seen in melanoma as well.
Depatuxizumab mafodotin (ADC, MMAF payload)	EGFR	EGFR-amplified glioblastoma	Phase 3 (INTELLANCE-1, NCT02573324)	No survival improvement when added to standard chemoradiation; Phase III trial in newly diagnosed GBM was negative, leading to termination of the program.
Ladiratuzumab vedotin (ADC, MMAE)	LIV-1	Triple-negative breast cancer; LIV-1–expressing solid tumors	Phase 1b/2 (NCT03310957)	Ongoing. Manageable toxicity and preliminary efficacy in TNBC. In first-line TNBC (PD-L1+), ladiratuzumab + pembrolizumab showed ~33% ORR, warranting further development.
Tasidotin (ILX-651; synthetic peptide)	Tubulin	Advanced solid tumors (refractory)	Phase 1	Dose-limiting neutropenia at higher doses. No dramatic responses; some anti-tumor activity with stable disease observed. Notably, melanoma CR reported, but no further development beyond Phase I.
Soblidotin (TZT-1027)	Tubulin	Non–small cell lung cancer (refractory NSCLC)	Phase 2	No objective responses in Phase II; median time to progression ~1.5 months. Showed minimal efficacy, and development was halted for NSCLC.
Lifastuzumab vedotin (DNIB0600A, ADC)	NaPi2b	Non-sq NSCLC; Platinum-resistant ovarian cancer	Phase 2 (NCT01991210)	LIFA achieved higher ORR (34% vs. 15%) than chemo, but PFS benefit was modest (5.3 vs. 3.1 mo) and not statistically significant. Consequently, the ADC was discontinued for lack of clear superiority.
PF-06263507 (anti-5T4 ADC)	5T4	Advanced solid tumors (5T4-expressing; lung, breast, ovarian)	Phase 1 (NCT01891669)	First-in-human dose escalation completed; ocular toxicities (e.g., photophobia, conjunctivitis) were dose-limiting. Demonstrated insufficient clinical activity; the program was discontinued after Phase I.
Telisotuzumab vedotin (“Teliso-V”, ADC)	c-MET	c-MET–overexpressing NSCLC (EGFR wild-type)	Phase 2 (LUMINOSITY, NCT03539536)	Achieved durable responses in c-MET high non-squamous NSCLC. In c-MET high patients, ORR ~53% with prolonged benefit. Received FDA accelerated approval in 2023 for advanced NSCLC with high c-MET expression. Common toxicities: fatigue, peripheral neuropathy (manageable).
Mecbotamab vedotin (BA3011, CAB-ADC)	AXL	Solid tumors with AXL expression	Phase 2 (NCT03425279)	Conditionally active ADC (pH-dependent tumor targeting). Interim Phase II results in refractory NSCLC show promising efficacy (objective responses in heavily pretreated patients). Development ongoing; well-tolerated with limited off-tumor toxicity due to CAB activation mechanism.
Ozuriftamab vedotin (BA3021, CAB-ADC)	ROR2	ROR2-positive cancers (melanoma; head and neck SCC; NSCLC)	Phase 2 (NCT03504488)	Conditionally active (CAB) ADC targeting ROR2. Early Phase II data in metastatic head and neck cancer showed encouraging response signals, leading to FDA Fast Track designation. Trials ongoing to confirm efficacy in ROR2-expressing tumors; toxicities so far manageable and mostly mild.

Abbreviations: ADC—antibody–drug conjugate; AXL—AXL receptor tyrosine kinase; CAB—conditionally active biologic; CR—complete response; EGFR—epidermal growth factor receptor; gpNMB—glycoprotein non-metastatic melanoma protein B; LIV-1—zinc transporter SLC39A6; MMAE/F—monomethyl auristatin E/F; NaPi2b—sodium-dependent phosphate transporter 2b; NSCLC—non-small-cell lung cancer; ORR—objective response rate; PFS—progression-free survival; PROC—platinum-resistant ovarian cancer; ROR2—receptor tyrosine kinase-like orphan receptor 2; TNBC—triple-negative breast cancer; 5T4—trophoblast glycoprotein (also known as TPBG); c-MET—hepatocyte growth factor receptor (HGFR).

## Data Availability

No new data were created or analyzed in this study. Data sharing is not applicable to this article.

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
