# Peer review of "Cyanobacterial Peptides in Anticancer Therapy: A Comprehensive Review of Mechanisms, Clinical Advances, and Biotechnological Innovation"

_marinedrugs, 2025, doi:10.3390/md23060233_

Round 1

Reviewer 1 Report

Comments and Suggestions for Authors

This manuscript is on the review of cyanobacterial peptides in anticancer therapy. The contents are well-organized. As such, I would recommend publication in Marine Drugs after the authors address the following minor point.

In Section 4.2 "Clinical-Stage Investigational Agents (Phases I-III)", the authors presented selected notable examples. A table listing additional candidates not covered in the manuscript is recommended to enhance the review's comprehensiveness.

Author Response

Thank you very much for taking the time to review this manuscript. Please find the detailed response below, and the corresponding revisions highlighted in the resubmitted files.

Comment: In Section 4.2 "Clinical-Stage Investigational Agents (Phases I-III)", the authors presented selected notable examples. A table listing additional candidates not covered in the manuscript is recommended to enhance the review's comprehensiveness.

Response: In response, we have expanded Section 4.2 and added a new Table 3 (pages 14–15, lines 540–553), which provides a broader overview of clinical-stage investigational agents employing cyanobacteria-inspired cytotoxic warheads. Specifically, we included several additional antibody–drug conjugates, such as lifastuzumab vedotin, telisotuzumab vedotin, PF-06263507, mecbotamab vedotin, and qzuriftamab vedotin. Although these agents were not initially discussed in the main text, their inclusion now serves to more comprehensively illustrate the continued translational relevance of cyanobacterial scaffolds in ADC-based oncology therapeutics. A corresponding explanatory sentence has also been added in Section 4.2 (page 13-14, lines 529–538) to clarify the rationale behind this addition:

To enhance the comprehensiveness of this section, Table 3 was added to provide an expanded overview of clinical-stage investigational agents inspired by cyanobacterial peptides. While several agents listed in Table 3 (e.g., glembatumumab vedotin, plitidepsin, OKI-179) are discussed in the text above, others—including lifastuzumab vedotin [94], PF-06263507 [95], telisotuzumab vedotin [96,97], mecbotamab vedotin [98,99], and ozuriftamab vedotin [100]—were included based on their use of auristatin payloads (MMAE or MMAF), which are synthetic analogs of dolastatin-10, a cyanobacterial peptide. These agents were not previously detailed in the main text but were incorporated to illustrate the broader clinical impact and structural versatility of cyanobacteria-derived cytotoxins in modern antibody–drug conjugate (ADC) development.

Table 3. Clinical-stage investigational agents derived from cyanobacterial peptides (Phases I–III, 2017–2025). This table expands the survey of candidates (beyond the approved agents in Table 2) to include additional peptide-based agents inspired by cyanobacteria, with their targets, trial status, and outcomes.

Agent (Type)

Target

Indications

Phase

Status/ Key Outcomes

Glembatumumab vedotin (ADC, MMAE payload)

gpNMB

Metastatic triple-negative breast cancer (TNBC); also tested in melanoma

Phase 2b

(METRIC, NCT01997333)

No benefit over chemo in TNBC (median PFS 2.9 vs 2.8 mo); significant toxicity (rash, neutropenia); discontinued after failing primary endpoint. Minimal activity seen in melanoma as well.

Depatuxizumab mafodotin (ADC, MMAF payload)

EGFR

EGFR-amplified glioblastoma

Phase 3 (INTELLANCE-1, NCT02573324)

No survival improvement when added to standard chemoradiation; Phase III trial in newly diagnosed GBM was negative, leading to termination of the program.

Ladiratuzumab

vedotin

(ADC, MMAE)

LIV-1

Triple-negative breast cancer; LIV-1–expressing solid tumors

Phase 1b/2 (NCT03310957)

Ongoing. Manageable toxicity and preliminary efficacy in TNBC. In first-line TNBC (PD-L1+), ladiratuzumab + pembrolizumab showed ~33% ORR, warranting further development.

Tasidotin (ILX-651; synthetic peptide)

Tubulin

Advanced solid tumors (refractory)

Phase 1

Dose-limiting neutropenia at higher doses. No dramatic responses; some anti-tumor activity with stable disease observed. Notably, melanoma CR reported, but no further development beyond Phase I.

Soblidotin

(TZT-1027)

Tubulin

Non–small cell lung cancer (refractory NSCLC)

Phase 2

No objective responses in Phase II; median time to progression ~1.5 months. Showed minimal efficacy, and development was halted for NSCLC.

Lifastuzumab

vedotin

(DNIB0600A, ADC)

NaPi2b

Non-sq NSCLC; Platinum-resistant ovarian cancer

Phase 2 (NCT01991210)

LIFA achieved higher ORR (34% vs 15%) than chemo, but PFS benefit was modest (5.3 vs 3.1 mo) and not statistically significant. Consequently, the ADC was discontinued for lack of clear superiority.

PF-06263507

(anti-5T4 ADC)

5T4

Advanced solid tumors (5T4-expressing; lung, breast, ovarian)

Phase 1 (NCT01891669)

First-in-human dose escalation completed; ocular toxicities (e.g. photophobia, conjunctivitis) were dose-limiting. Demonstrated insufficient clinical activity; the program was discontinued after Phase I.

Telisotuzumab

vedotin

(“Teliso-V”, ADC)

c-MET

c-MET–overexpressing NSCLC (EGFR wild-type)

Phase 2 (LUMINOSITY, NCT03539536)

Achieved durable responses in c-MET high non-squamous NSCLC.In c-MET high patients, ORR ~53% with prolonged benefit. Received FDA accelerated approval in 2023 for advanced NSCLC with high c-MET expression. Common toxicities: fatigue, peripheral neuropathy (manageable).

Mecbotamab

vedotin (BA3011, CAB-ADC)

AXL

Solid tumors with AXL expression

Phase 2 (NCT03425279)

Conditionally active ADC (pH-dependent tumor targeting). Interim Phase II results in refractory NSCLC show promising efficacy (objective responses in heavily pretreated patients). Development ongoing; well-tolerated with limited off-tumor toxicity due to CAB activation mechanism.

Ozuriftamab

vedotin (BA3021, CAB-ADC)

ROR2

ROR2-positive cancers (melanoma; head & neck SCC; NSCLC)

Phase 2 (NCT03504488)

Conditionally active (CAB) ADC targeting ROR2. Early Phase II data in metastatic head & neck cancer showed encouraging response signals, leading to FDA Fast Track designation . Trials ongoing to confirm efficacy in ROR2-expressing tumors; toxicities so far manageable and mostly mild.

*Abbreviations: ADC – antibody–drug conjugate; AXL – AXL receptor tyrosine kinase; CAB – conditionally active biologic; CR – complete response; DLT – dose-limiting toxicity; EGFR – epidermal growth factor receptor; gpNMB – glycoprotein non-metastatic melanoma protein B; LIV-1 – zinc transporter SLC39A6; MMAE/F – monomethyl auristatin E/F; NaPi2b – sodium-dependent phosphate transporter 2b; NSCLC – non–small cell lung cancer; ORR – objective response rate; PFS – progression-free survival; PROC – platinum-resistant ovarian cancer; ROR2 – receptor tyrosine kinase-like orphan receptor 2; TNBC – triple-negative breast cancer; 5T4 – trophoblast glycoprotein (also known as TPBG); c-MET – hepatocyte growth factor receptor (HGFR).

Reviewer 2 Report

Comments and Suggestions for Authors

The review is devoted to the prospects of using cyanobacterial peptides, mainly cyclopeptides and depsipeptides, as anticancer agents, and also touches on the issues of biotechnological production of these peptides, increasing their targeting and possible molecular mechanisms of action. This topic is very relevant due to the increase in oncological diseases. The presentation of the material is well crafted and the text itself is written in excellent language.

Author Response

We sincerely thank for the thoughtful and encouraging feedback. We are pleased to hear that the topic and structure of our review were well received, and we deeply appreciate your positive evaluation of the manuscript’s clarity and language quality. Your comments reinforce the importance of continued exploration in this area and encourage us in our ongoing research efforts.

Reviewer 3 Report

Comments and Suggestions for Authors

After carefully reviewing the manuscript “Cyanobacterial Peptides in Anticancer Therapy: A Comprehensive Review of Mechanisms, Clinical Advances, and Biotechnological Innovation,” I believe it can be published in its current form.

-The topic is hot, and it might be interesting to many readers.

-The authors designed the manuscript very well, and the presented data are clear and transparent.

-The authors reviewed comprehensively all the aspects of using cyanobacterial peptides for the treatment of some cancers.     

Author Response

We sincerely thank for the positive and encouraging feedback. We are truly grateful for your recognition of the manuscript’s clarity, design, and comprehensive coverage of cyanobacterial peptides in anticancer therapy. Your supportive comments are highly appreciated and motivating for our team.

Reviewer 4 Report

Comments and Suggestions for Authors

The manuscript is well structured and organized, namely the information sistematized in tables 1 and 2 and the figures illustrating action mechanisms and biotechnological approaches for biotechnological production. The information collected herein is relevant and extremely used for researchers in the field of anticancer drug discovery.

Author Response

We sincerely thank for the positive and encouraging feedback. We are grateful that the structure and organization of the manuscript, particularly the summarized data in Tables 1 and 2 and the explanatory figures, were found to be effective. It is truly rewarding to know that the content presented may be of value to researchers in the field of anticancer drug discovery. Your comments are highly appreciated and motivate us to continue advancing research in this area.